# Identification and Typing of Strains of Wood-Rotting Basidiomycetes by Protein Profiling Using MALDI-TOF MS

**DOI:** 10.3390/biotech11030030

**Published:** 2022-07-27

**Authors:** Sakae Horisawa, Koki Iwamoto

**Affiliations:** Department of Environmental Systems Engineering, Faculty of Engineering, Kochi University of Technology, Kami-gun, Kochi 782-8502, Japan; kouki-iwamoto@yamakin-gold.co.jp

**Keywords:** MALDI-TOF MS, MALDI Biotyper, basidiomycetes, species identification, fingerprinting

## Abstract

The accurate identification and proper typing of basidiomycetes are required in medical, sanitary maintenance, agriculture, and biotechnology fields. A diagnostic method based on information from whole-cell proteins acquired by matrix-assisted laser desorption/ionization time-of-flight mass spectrometry (MALDI-TOF MS) was investigated to identify wood-rotting fungi, a group of filamentous fungi. In this study, mass spectra of intracellular peptides obtained from cultured mycelia of 50 strains of 10 wood-rotting fungal species were obtained multiple times and mass spectral patterns (MSPs) consisting of peaks that characterized the fungal species or strain was created to construct an in-house database. The species identification was conducted by comparing the newly obtained raw mass spectra with the MSPs in the database using the MALDI Biotyper. The results showed that the peak patterns of the mass spectra were reproducible and matched at the strain level. A cluster analysis based on the MSPs was also conducted to examine inter-and intraspecific diversity among the tested wood-rotting basidiomycetes. Most of the fungal strains examined in this study could be identified to a species level; however, the strains belonging to *Pleurotus* could only be identified to a genus level. This was due to an intraspecific variation, so the identification accuracy could be amendable with a more enhanced database.

## 1. Introduction

The accurate identification of filamentous fungi is important for microbiological applications, medical-related issues, sanitary supervision, and the agricultural industry. Previously, filamentous fungi were mainly identified based on morphological observations and physiological characteristics such as the growth environment and nutrient requirements; however, recent methods using biomolecules that do not rely on the experience of the examiner are now frequently used. In particular, methods based on DNA sequences have been highly developed with rapid progress [1,2]. Nucleotide sequences of ribosomal DNA are frequently used because they have both conservative regions for the easy acquisition of PCR products from unknown organisms and variable regions to detect species-specific variations. However, analyses of intracellular proteins have been used for phylogenetic analyses and typing at a wide range of taxonomic levels [3,4].

Matrix-assisted laser desorption/ionization time-of-flight mass spectrometry (MALDI-TOF MS) is a rapid and low-cost analysis method for proteins that can identify organisms at species or genus levels [5,6]. Species identification methods using MALDI-TOF MS can identify or detect inter- and intraspecific variations by acquiring the spectra of all relatively small peptides of approximately 2000–20,000 Da in a cell and comparing them with the spectra of known bacterial species. The main peaks of these mass spectra are of a ribosomal protein origin, accounting for approximately 50–70% of the mass spectra [7,8]. Mass spectral databases have been established mainly for bacteria and yeasts related to clinical medicine. It was reported that the results of a spectral analysis were consistent with the results of a DNA analysis at a high rate, indicating that the method has practical applications [9,10,11]. The mass spectra represent ‘fingerprint’ information specific to each microbial strain; therefore, it can be applied to infer pathogenicity or drug susceptibility within a species or the classification of brewer’s yeasts [12,13,14,15].

The identification and diagnosis of filamentous fungi by MALDI-TOF MS are mainly limited by the low coverage of filamentous fungi in commercial databases and the time-consuming pretreatment of mold samples to obtain a sufficient quality of the mass spectra because molds have stronger cell walls than bacteria [16,17]. Mass spectral databases are being created for disease/clinical-related filamentous fungi [18,19,20,21]. The species identification and typing of fungal strains of the plant symbiotic *Glomeromycota* using protein profiling extracted from the spores by MALDI-TOF MS have also been reported [22].

Basidiomycetes, examined in the present study, decompose various types of organic matter in nature and play essential roles in the material cycle such as supplying nutrients in the soil to plants. Wood-rotting basidiomycetes are industrially and socially important because they are involved in wood decomposition and used as components of buildings; additionally, a few groups are agricultural food products [23]. Basidiomycetes have mycelia similar to ascomycetes, but their life cycles slightly differ. They have a few unique characteristics; for example, the sexual spores are produced from special cells called basidia. Wood-rotting basidiomycetes can degrade plant cell wall components, cellulose, hemicellulose, and lignin. Therefore, they are industrially important because they cause the deterioration of the wood structure. However, there are still few reports on the identification and typing of basidiomycetes using MALDI-TOF MS [24,25]. The present study investigated the identification and typing of wood-rotting fungi using MALDI-TOF MS.

## 2. Materials and Methods

### 2.1. Fungal Strains

A total of 10 major species of wood-rotting fungi (50 strains) were examined in this experiment (Table 1). Each strain was sub-cultured on a potato dextrose agar (PDA) medium at 25 °C in the dark. For all strains, the taxonomic species was confirmed based on the nucleotide sequence of the internal transcribed spacer (ITS) region and a part of large subunit of ribosomal rDNA (D1/D2). The ITS region sequences of the strains collected from GenBank/DDBJ/EMBL-Bank were aligned and a dendrogram was generated using MEGA version 11 [26] to visualize the inter- and intraspecific variation. To determine the discriminatory ability of the ITS gene sequencing and MALDI-TOF MS, the pairwise distances were calculated and displayed in a tree generated by the unweighted pair group method with arithmetic averaging using MEGA version 11.

### 2.2. Protein Extraction

The protein of each strain was extracted from colonies with a >4 cm radius on the PDA that reached the stationary phase. The fungal proteins were extracted using the ethanol–formic acid extraction method [27,28]. The mycelia cultured on the PDA plates were collected and placed in 1.5 mL tubes. Subsequently, 800 µL of MilliQ water was added to the tube to wash the mycelia. The tube was then centrifuged to separate the mycelial pellets for the supernatant removal. A total of 900 µL ethanol was then added to the pellet to prevent the protein deterioration and the supernatant was removed after centrifugation. After the ethanol was completely removed by further decompression drying, the mycelia were transferred to a 2.0 mL tube with 100 µL of 70% formic acid and zirconia beads (a bead with a 5 mm diameter and an aliquot of 0.2 mm beads). The tube was shaken using a Tissue Lyser (QIAGEN, Venlo, The Netherlands) for 1.5 min at 25 Hz to crush the mycelium. Finally, 100 µL of acetonitrile was added to the tube and centrifuged to separate the liquid and pellets. The supernatant in the tube was collected as the protein extraction solution.

### 2.3. MALDI-TOF MS Analysis

The extracted protein was analyzed using MALDI-TOF MS to acquire the mass spectra of the whole-cell protein from the basidiomycetes tested in this study. The mass spectra of the test fungi were observed using Autoflex Speed Instrument (Bruker, Bremen, Germany). α-Cyano-4-hydroxycinnamic acid (CHCA) (Bruker) was employed as a matrix. First, 100 µL of an organic solvent (OS) including 50% acetonitrile and 2.5% trifluoroacetic acid was added to 100 mg of CHCA and repeatedly stirred by a vortex mixer to completely dissolve the CHCA crystals. The matrix solution was stored in the dark (4 °C) until use. The Bruker Bacterial Test Standard (BTS, Bruker) was used as the calibration standard. A total of 50 µL of OS was added to the 5 µL of BTS and thoroughly mixed by pipetting; 10 µL of the supernatant was dispensed into 0.5 mL tubes and stored at −28 °C until use.

Subsequently, 1 µL of the protein solution was spotted onto a MALDI sample target plate (MTP target frame III, Bruker) and dried at room temperature. An amount of 1 µL of the matrix solution was placed on each protein spot to co-crystallize with the proteins. The MALDI-TOF mass spectra were observed using an Autoflex Speed Instrument (Bruker) operated in a linear positive mode with Flex Control version 3.0. Positive ions were extracted with an accelerating voltage of 20 kV in a linear mode. For each sample, at least eight mass spectra were collected through at least three repeat measurements; each measurement consisted of five spots with at least four shots each. The spectra were analyzed in the 2000–20,000 *m*/*z* range.

### 2.4. MALDI-TOF MS Data Interpretation

The mass spectra were baseline-corrected and smoothing-processed with Biotyper version 3.0 software (Bruker Daltonics, Bremen, Germany). Mass spectral patterns (MSPs) were created from the spectra that were acquired multiple times for each strain by selecting the main peaks and removing very minor peaks and noise using the MALDI Biotyper. An in-house MSP database of wood-rotting fungi was constructed. The identification scores of the newly acquired mass spectra of the MSPs in the database were calculated by the MALDI Biotyper to evaluate the reproducibility. The scores were presented between 0 and 3: a score ≥ 2.0 was accepted for a reliable identification to the species level; a score ≥ 1.7 and < 2.0 was accepted for an identification to the genus level; and a score < 1.7 indicated an unreliable identification of the microorganisms [29,30].

## 3. Results

### 3.1. Comparison of Inter- and Intraspecific MS Patterns

The MSPs generated from the multiple raw mass spectra for each fungal strain are shown in Figure 1. The incubation periods of the strains *Bjerkandera*
*adusta* NBRC 4928, *Fomitopsis palustris* NBRC 30339, *Pleurotus ostreatus* NBRC 30160, *P. pulmonarius* NBRC 31345, *P. eryngii* PeCM, *P. citrinopileatus* NBRC 30528, *Schizophyllum commune* NBRC 30496, *Serpula lacrymans* HFP 7906, *Trametes versicolor* NBRC 30340, and *T. hirsuta* NBRC 4917 were 9, 6, 7, 9, 14, 12, 10, 10, 18, 8, and 9 days, respectively. The vertical axis showed the relative values, but the axis scales of *P. ostreatus, P. eryngii*, and *P. pulmonarius* were adjusted because of the large difference in the peak values. Peaks observed around 3000–3500, 6000–7000, and 9000 *m*/*z* were believed to characterize the fungal species.

The comparisons of the MSPs within each species or genus are shown in Appendix A. The spectra of *B. adusta, F. palustris*, and *S. commune* had a slightly greater variation in each species whereas those of *P. ostreatus, S. lacrymans, T. versicolor,* and *T. hirsuta* were similar. A dendrogram for the comparison of the MSPs of all tested fungal strains was generated using MALDI Biotyper version 3.0 (Figure 2). The dendrogram showed that all the species created distinct and separate clusters, except for the genus *Pleurotus*; *T. versicolor* and *T. hirsuta* were completely separated even though the distance in each species was larger (the branches were longer). However, for the genus *Pleurotus*, *P. citrinopileatus* belonged to a distinct cluster, but *P. eryngii, P. ostreatus,* and *P. pulmonarius* belonged to the same cluster. A dendrogram based on the ITS region sequence showed that the cluster of each species was independently formed, except for the genus *Pleurotus*; there was little variation in a species (Figure 3). The cluster formation in the species was similar between the two dendrograms.

### 3.2. Examination of Mass Spectral Variation Using the MALDI Biotyper Score

The scores between the newly acquired raw mass spectra and the MSPs in the in-house database were calculated using the MALDI Biotyper to investigate the reproducibility of the mass spectra of the strains and the inter- and intraspecific diversity for the genera *Bjerkandera*, *Pleurotus*, and *Trametes*. The average and standard deviation of the scores for each strain are shown in Figure 4, Figure 5 and Figure 6. Good spectral reproducibility was inferred if the MSP score of each strain and the newly obtained spectrum of the same strain was >2.0.

In the genus *Bjerkandera*, the strains were divided into two groups according to the score; the group including IWA5b, FERM P-20326, and NBRC 106826 had a little variation and the other had a substantial variation (Figure 4). This finding was consistent with the *Bjerkandera* cluster formed in the dendrogram in Figure 2. For bacteria, the Biotyper scores of ≥2.0 indicate species-level identification whereas scores between 1.7 and 2.0 indicate genus-level identification. For the wood-rotting basidiomycetes in this study, scores > 2.0 were not seen, even at the species level. This suggests that identification at the intraspecific level, i.e., at the strain level, is possible. There were two groups in *P. ostreatus*; the strains NBRC 6515, 30160, 30880, and 104981 showed similar mass spectral patterns whereas NBRC 8330 and 33211 were more variable (Figure 5). NBRC 8330 and 33211 showed lower scores compared with other *P. ostreatus* strains, even though they were classified as the same species. The score of each strain in *P. pulmonarius* and *P. citrinopileatus* was ≥2.0 only for the MSPs of the same strain. In the genus *Trametes*, the intraspecific variation was found to be lower in *T. versicolor* and higher in *T. hirsuta*, but both species could be completely identified to a species level based on the Biotyper score (Figure 6).

### 3.3. Consideration of Strain Classification

The two dendrograms in this study indicated that there was a need to discuss the classification of the two *Pleurotus* strains described above. The MSP of NBRC 8330, registered as *P. ostreatus* in the Japanese strain collection center NBRC, placed this strain in the *P. pulmonarius* cluster of the dendrogram (Figure 2). Another cluster analysis based on the ITS DNA sequences also showed that NBRC 8330 was placed in the *P. pulmonarius* cluster (Figure 3). The BLAST search results from GenBank/DDBJ/EMBL-Bank showed that the ITS DNA sequence of NBRC 8330 showed a high homology with both *P. pulmonarius* and *P. ostreatus*. NBRC 33211 was also registered in the NBRC collection as *P. ostreatus*; however, a cluster analysis of the MSPs showed it was not in the *P. ostreatus* cluster, which had a little variation, but in the *P. eryngii* cluster. Furthermore, an examination within the genus *Pleurotus* based on the mass spectral information (Figure 5) indicated that NBRC 8330 was closely related to *P. pulmonarius* NBRC 31345. Similarly, NBRC 33211 was shown to be closely related to *P. eryngii*. It has been suggested that these strains may be in a phylogenetically intermediate position of these species, respectively, and the analysis using mass spectra could represent that.

## 4. Discussion

The advantages of using MALDI-TOF MS spectra to identify unicellular microorganisms such as bacteria and yeasts include a simple pretreatment with a small number of bacteria [6], an insignificant loss of ionization efficiency even if the protein is not purified, and ease of spectral analysis because of the predominant production of univalent ions [31,32,33]. For unicellular microorganisms, a simple sample preparation method can be applied in which the bacteria are directly applied to a steel plate and overlaid with a matrix to simultaneously extract and crystallize the protein [34]. However, prior extraction was necessary for the filamentous fungi used in this study because of their rigid cell walls. Thus, the crude proteins extracted contained other biomolecules such as lipids, which may provide a more detailed profile of the fungus (fingerprinting) [34].

Basidiomycetes have not accumulated sufficient whole-genome data to select marker protein peaks for characterizing the fungal species in mass spectra using bioinformatics. Therefore, the fingerprinting method [6] is considered suitable for basidiomycete identification and typing of strains, but protein extraction and pre-culturing conditions must be strictly controlled to improve its accuracy. As shown in Figure 1, it was clear that the strains examined in this study were almost identifiable. The mass spectral pattern of each strain was characterized by a molecular weight peak of approximately 2000–10,000 Da, which was similar to that of bacteria and yeasts that have already been reported [15,35,36]. Most of the *F. palustris* strains studied here were isolated in Japan (except NBRC 30339) [37]; therefore, we expected their mass spectra to be highly similar. However, it was possible that there might be two groups within the same species based on the mass spectrum pattern. The 12 *S. lacrymans* strains tested were determined to have a low intraspecific variation by the mass spectral comparison, which was consistent with the results of genomic information-based typing using an RAPD analysis [38]. The results indicated that the typing based on the mass spectra was able to identify an intraspecific variation that could not be detected by DNA barcode sequences such as ITS regions. The mass spectra of four strains of *S. commune* acquired in the present study were compared with the information on *S. commune* in the commercial database in Biotyper 3.0; the scores ranged from 1.0 to 1.7. The reason for that could be due to a variation within the species, culture conditions, or other factors.

To enhance the mass spectra database, intra- and interspecific variations should be examined in all fungal species. In the present study, mass spectra variations in three genera, *Bjerkandera*, *Trametes*, and *Pleurotus*, were investigated because each genus exhibited characteristic features of intra- and interspecific variations in the mass spectra: the genus *Bjerkandera* had two subgroups; in the genus *Pleurotus*, the mass spectral analysis detected taxonomically ambiguous strains by DNA barcoding and suggested a taxonomic position for the strains; and in the genus *Trametes*, the mass spectra information could be clearly separated into species. The investigation suggested that a mass spectra comparison is an effective tool to identify and discriminate wood-rotting fungi.

In this study, the fungal spectral patterns acquired by the analysis of whole-cell proteins extracted from mycelia cultured on PDA showed sufficient species specificity to diagnose the fungal species, except for the genus *Pleurotus*, for which only genus-level identification was possible. In general, fungi such as basidiomycetes and ascomycetes have various life stages such as mycelia, fruiting bodies, and spores. In ascomycetes that frequently form spores, information from the proteins of the spores can be used for identification [39]. In basidiomycetes, spores are mainly produced on the fruiting bodies; therefore, it seems adequate to use those of mycelial proteins. The fungal spectra, including those of basidiomycetes, probably change based on the type of media and the age of the mycelia. Mass spectra reproducibility could be improved by synchronizing the experimental conditions, including the incubation period [40].

The identification of unknown filamentous fungi using mass spectra requires isolated strains. Once they are in hand, the MS analysis can be performed in a very short time and at a low cost, which surpasses the method using DNA barcoding. This method using mass spectra could be of great help in clinical applications and in the determination of other pathogens. For basidiomycetes, there is a problem of a large intraspecific variation in the mass spectra—as the present study showed—so it is essential to enhance the database for use as an identification tool; it is believed that an enhanced database will allow for a more accurate diagnosis of the fungi. As an example of a suitable application, this approach has great potential for managing strain collections in laboratories and institutes that store many microorganism strains; this approach would be superior to a DNA analysis because it is inexpensive and rapid. As the fingerprinting method allows a detailed characterization within a species, mass spectral identification is expected to be useful for strain-level typing such as detecting intraspecific variations and drug resistance.

## 5. Conclusions

The MSPs obtained from the MALDI-TOF MS spectra of proteins extracted from fungal mycelia were examined for the identification and typing of wood-rotting basidiomycetes. The MSP of each strain was found to have a high species or strain specificity and it seems suitable for its application to basidiomycetes for species determination and the maintenance of fungal collections. For more accurate biotyping, it is necessary to examine the MSP variations caused by the culture conditions and culture stage.

## Figures and Tables

**Figure 1 biotech-11-00030-f001:**
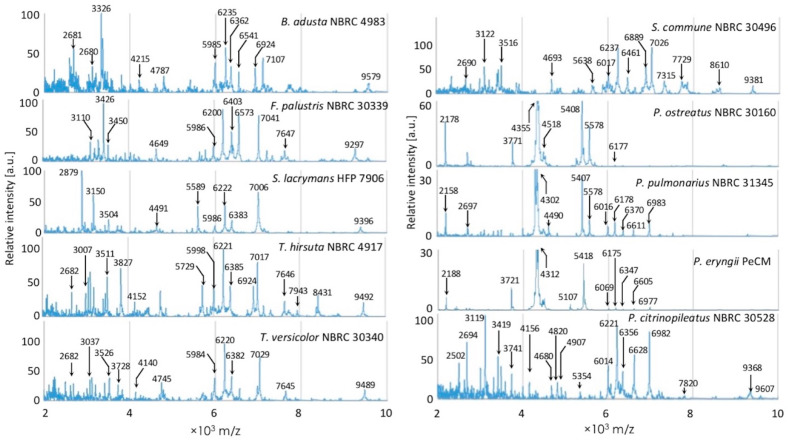
Raw mass spectra of representative strains of each species. *Bjerkandera adusta*, NBRC 4983; *Fomitopsis palustris*, NBRC 30339; *Serpula lacrymans*, HFP 7906; *Trametes hirsuta*, NBRC 4917; *Trametes versicolor*, NBRC 30340; *Schizophyllum commune*, NBRC 30496; *Pleurotus ostreatus*, NBRC 30160; *Pleurotus pulmonarius*, NBRC 31345; *Pleurotus citrinopileatus*, NBRC 30528. The y-axis indicates relative intensity of the recorded peaks; the x-axis shows the mass changes from 2000 to 10,000 Da.

**Figure 2 biotech-11-00030-f002:**
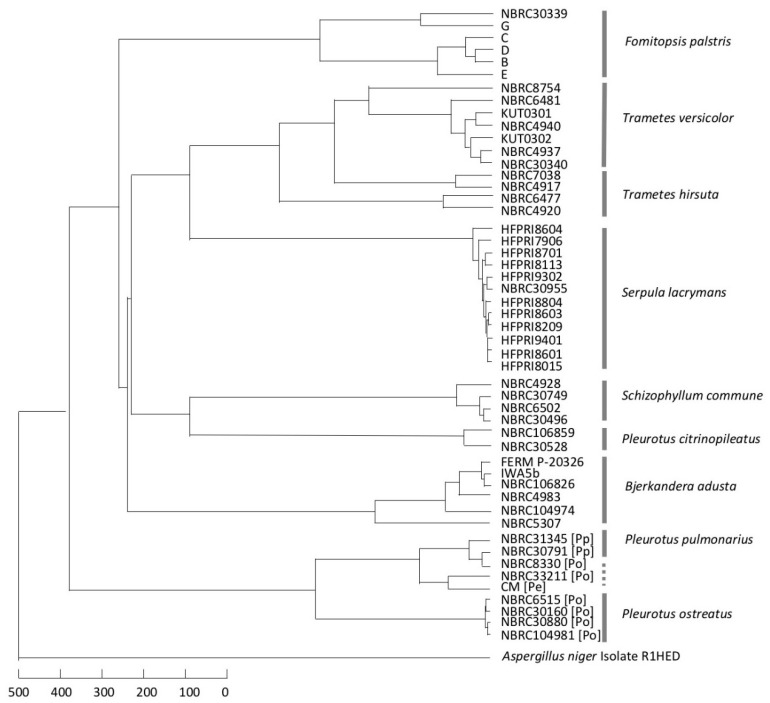
Mass spectral pattern (MSP) dendrogram of extracted whole-cell proteins from wood-rotting fungi. Pp: *P. citrinopileatus*; Po: *P. ostreatus*; Pe: *P. eryngii*; Pc: *P. citrinopileatus*; Th: *T. hirsuta*; Tv: *T. versicolor*. Mass spectrum information of *Aspergillus niger* in the Biotyper version 3.0 database was employed as an outgroup. The dendrogram was created using Biotyper and distance units correspond with the relative similarity of MSPs.

**Figure 3 biotech-11-00030-f003:**
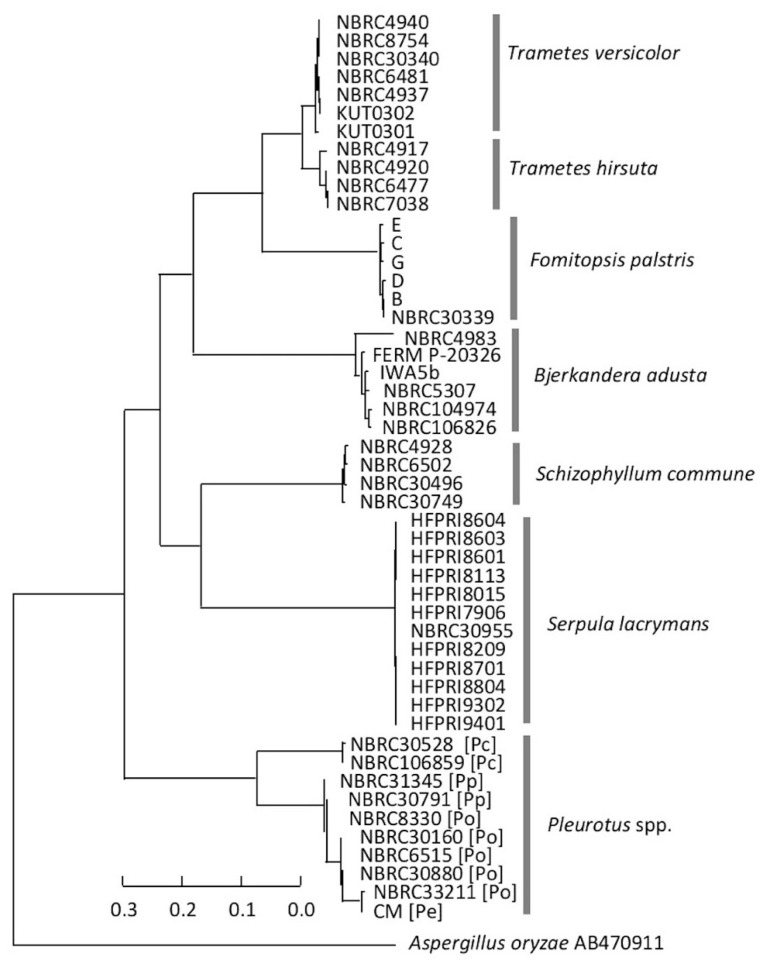
MSP dendrogram based on the nucleotide sequence of the internal transcribed spacer region from wood-rotting fungi. The sequence distances are given as a percent difference. Pp: *P. citrinopileatus*; Po: *P. ostreatus*; Pe: *P. eryngii*; Pc: *P. citrinopileatus*; Th: *T. hirsuta*; Tv: *T. versicolor.* The dendrogram was created using Biotyper version 3.0 and distance units correspond with the relative similarity of MSPs.

**Figure 4 biotech-11-00030-f004:**
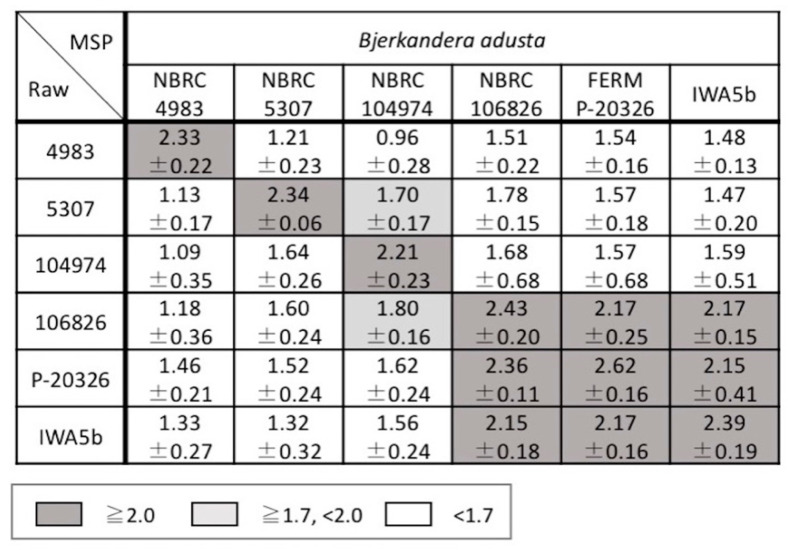
Average identification scores of raw mass spectra and the MSPs of *Bjerkandera adusta*. The scores presented are between 0 and 3: ≥2.0 (gray cells) was accepted for reliable identification to the species level; ≥1.7 and <2.0 (light gray cells) was accepted for identification to the genus level; and <1.7 (white cells) indicated unreliable identification of fungi.

**Figure 5 biotech-11-00030-f005:**
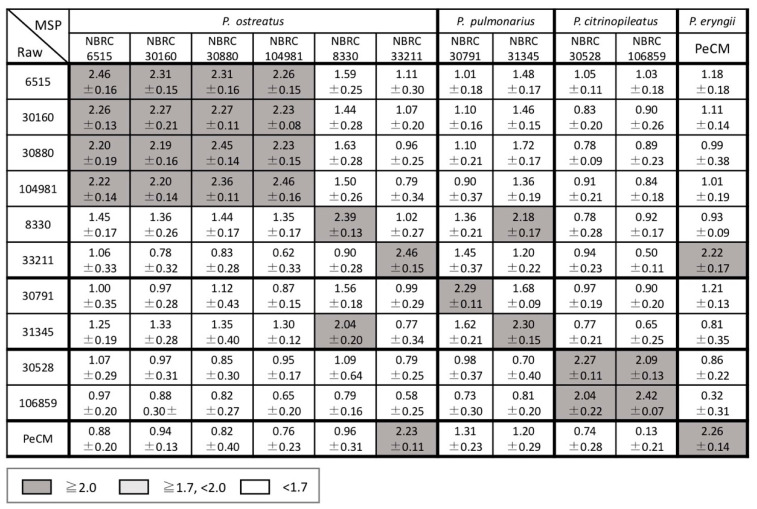
Average identification scores of raw mass spectra and the MSPs of *Pleurotus*. The scores presented are between 0 and 3: ≥2.0 (gray cells) was accepted for reliable identification to the species level; ≥1.7 and <2.0 (light gray cells) was accepted for identification to the genus level; and <1.7 (white cells) indicated unreliable identification of fungi.

**Figure 6 biotech-11-00030-f006:**
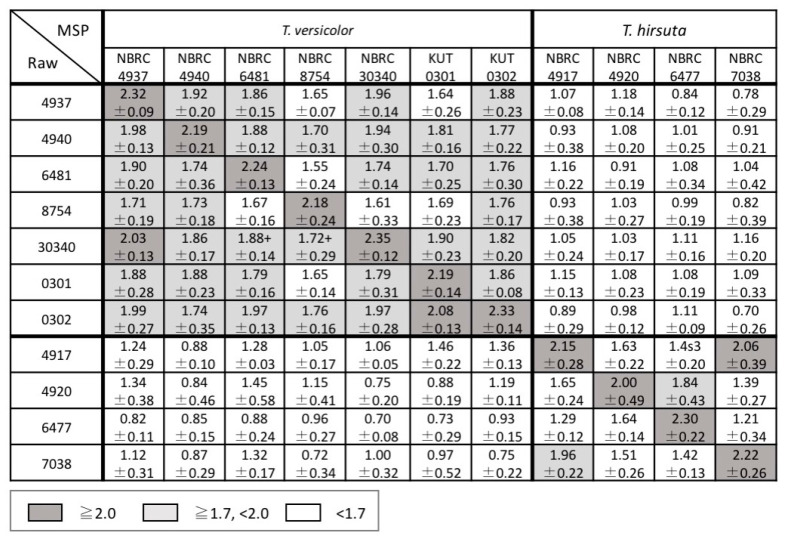
Average identification scores of raw mass spectra and the MSPs of *Trametes*. The scores presented are between 0 and 3: ≥2.0 (gray cells) was accepted for reliable identification to the species level; ≥1.7 and <2.0 (light gray cells) was accepted for identification to the genus level; and <1.7 (white cells) indicated unreliable identification of fungi.

**Table 1 biotech-11-00030-t001:** List of analyzed fungal strains.

Scientific Name	Strain Name	Locality of Source	Acc. No. of ITS and D1D2 Sequences
** *Bjerkandera adusta* **	NBRC 4983	-	AB733156, AB733333
	NBRC 5307	-	AB592333, LC705002
	NBRC 104974	Kyoto, Japan	AB733157, AB733334
	NBRC 106826	Hokkaido, Japan	LC705000, AB733334
	FERM P-20326	Japan	LC704999, -
	IWA5b	Kochi, Japan	LC705001, -
** *Fomitopsis palustris* **	NBRC 30339	-	AB733120, AB733302
	NarF-B	Kagoshima, Japan	LC705003, -
	NarF-C	Kagoshima, Japan	LC705004, -
	NarF-D	Kagoshima, Japan	LC705005, -
	NarF-E	Miyagi, Japan	LC705006, -
	NarF-G	Gifu, Japan	LC705007, -
** *Pleurotus citrinopileatus* **	NBRC 30528	Aomori, Japan	LC713431 *
	NBRC	Hokkaido, Japan	LC713434 *
** *Pleurotus eryngii* **	PeCM	-	LC713435 *
** *Pleurotus ostreatus* **	NBRC 6515	-	AB733142, AB733315
	NBRC 8330	Japan	LC713430 *
	NBRC 30160	Hyogo, Japan	AB733143, AB733316
	NBRC 30880	Korea	LC713432 *
	NBRC 33211	Kyoto, Japan	AB733144, AB733317
	NBRC 104981	Kyoto, Japan	LC713433 *
** *Pleurotus pulmonarius* **	NBRC 30791	India	AB733145, AB733318
	NBRC 31345	Tottori, Japan	LC705008, AB733319
** *Schizophyllum commune* **	NBRC 4928	-	AB733163, AB733339
	NBRC 6502	-	AB733164, AB733340
	NBRC 30496	Osaka, Japan	AB733165, AB733341
	NBRC 30749	Osaka, Japan	AB733166, AB733342
** *Serpula lacrymans* **	NBRC 30955	Hokkaido, Japan	AB733149, AB733324
	HFPRI 7096	Hokkaido, Japan	AB733420, AB733411
	HFPRI 8015	Hokkaido, Japan	AB733421, AB733412
	HFPRI 8113	Hokkaido, Japan	AB733422, AB733413
	HFPRI 8209	Hokkaido, Japan	AB733423, AB733414
	HFPRI 8601	Hokkaido, Japan	AB733424, AB733415
	HFPRI 8603	Nagano, Japan	AB733426, AB733416
	HFPRI 8604	Nagano, Japan	AB733427, AB733417
	HFPRI 8701	Toyama, Japan	AB733428, AB733418
	HFPRI 8804	Hokkaido, Japan	AB733429, AB733348
	HFPRI 9302	Hokkaido, Japan	AB733430, AB733347
	HFPRI 9401	Tokyo, Japan	AB733431, AB733419
** *Trametes hirsuta* **	NBRC 4917	-	AB733167, AB733343
	NBRC 4920	-	AB733169, AB733344
	NBRC 6477	-	AB733168, AB733345
	NBRC 7038	-	AB733170, AB733346
** *Trametes versicolor* **	NBRC 4937	-	AB733152, AB733328
	NBRC 4940	-	AB733153, AB733329
	NBRC 6481	-	LC705011, AB733330
	NBRC 8754	-	AB733154, AB733331
	NBRC 30340	-	AB733151, AB733327
	KUT 0301	Kochi, Japan	LC705009, -
	KUT 0302	Kochi, Japan	LC705010, -

NBRC: National Biological Resource Center of Japan; *: DNA sequence including ITS and D1D2 sequences.

## Data Availability

Not applicable.

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
