# Peer review of "Identification and Typing of Strains of Wood-Rotting Basidiomycetes by Protein Profiling Using MALDI-TOF MS"

_biotech, 2022, doi:10.3390/biotech11030030_

Round 1

Reviewer 1 Report

The manuscript analyzes the identification of wood-rotting basidiomycetes by MALDI-TOF MS. There are few papers about application of MALDI-TOF MS on these fungi, so I enjoyed reading this study. The study is technically well conducted. Nevertheless, I found some lacking information that I would like to know, and they will improve the paper. Several revisions are required.

Major comments:

1. Line 50. The authors are talking about brewer's yeast. However, references 12 and 13 are papers of mycobacteria. Please consider changing these two references and provide more accurate ones.  

2. Regarding references 16-21, I found them quite outdated. There have been a lot of research on fungi and MALDI-TOF in last years. Although some of them can be maintained due to their importance (Croxatto A et al, Schulthess B et al.), I reccomend to update some. For example:
https://doi.org/10.1093/mmy/myx154
https://doi.org/10.3389/fmicb.2019.02098
https://doi.org/10.1128/jcm.01299-21

3. Lines 67-68. The authors says that there are few reports on the identification of basidiomycetes by MALDI-TOF, but there are no cited references here. It is important to include them in order to know and provide to the reader the other studies on the same field. For example these:
About Amanita: https://doi.org/10.1093/mmy/myab018
About wild mushrooms: https://doi.org/10.1016/j.aca.2016.05.056

4. Materials and methods, lines 115-116. It is no clear how many spots have been used per strains and how many reads have been performed on each spot. 

5. In section 2.4: Have the authors used FlexAnalysis program to smoothing and baseline substraction of the spectra, or other analyses? If so, it should be mentioned. 

6. Schizophyllum commune is already in the commercial database of Bruker. Have you evaluated the possibility of identify it with the manufacturer's database?

7. The authors have evaluated the scores between raw spectra and MSP in the database for Bjerkandera, Pleurotus and Trametes. Why you have not included the other genera (Fomitopsis, Serpula, Schizophyllum)? It should be confirmed that they are also properly identified and no misidentifications at genus level are obtained. 

Minor comments:

1. Materials and methods: which was the approximate time that the cultures were incubated?

2. Figure 4, 5 and 6. In the legend, I recommend to change the score "≥1.7,<2.0" to "1.7-2.0" to be clearer. In addition, this box is lacking in the legend of Figure 5. 

3. Reference 33 have no title. Please, correct it. 

Author Response

Dear Reviewer 1,

We thank the reviewers1 for carefully reading our manuscript and providing numerous kind and helpful comments. We have revised the manuscript according to the reviewers' comments.

Major comments:

  1. Line 50. The authors are talking about brewer's yeast. However, references 12 and 13 are papers on mycobacteria. Please consider changing these two references and provide more accurate ones.  
  • Thank you for providing pointing this out. These were not appropriate. We have replaced these with new references recommended.
    1. Elsa Gorre, Cathy Muste & Kevin G. Owens (2018) Introducing a Cell-Free Approach for the Identification of Brewing Yeast (Saccharomyces cerevisiae) Strains Using MALDI-TOF MS. Journal of The American Society for Mass Spectrometry 29, pages2248–2259. https://doi.org/10.1007/s13361-018-2031-x
    2. Wieme AD, Freek Spitaels, Peter Vandamme, Anita Van Landschoot (2014) Application of matrix-assisted laser desorption/ionization time-of-flight mass spectrometry as a monitoring tool for in-house brewer's yeast contamination: a proof of concept. J. Inst. Brew. 120(4), pp. 438-443https://doi.org/10.1002/jib.149

2. Regarding references 16-21, I found them quite outdated. There have been a lot of research on fungi and MALDI-TOF in last years. Although some of them can be maintained due to their importance (Croxatto A et al, Schulthess B et al.), I reccomend to update some. For example:
https://doi.org/10.1093/mmy/myx154
https://doi.org/10.3389/fmicb.2019.02098
https://doi.org/10.1128/jcm.01299-21

  • Thank you for your suggestions. All the reports recommended were very interesting. We regret our lack of study. We have replaced these older references with the recommended papers except for the one (Croxatto A et al., Schulthess B et al.).
  1. Lines 67-68. The authors says that there are few reports on the identification of basidiomycetes by MALDI-TOF, but there are no cited references here. It is important to include them in order to know and provide to the reader the other studies on the same field. For example these:
    About Amanita: https://doi.org/10.1093/mmy/myab018
    About wild mushrooms: https://doi.org/10.1016/j.aca.2016.05.056
  • Thank you for your suggestions. These were also interesting papers. We have cited the papers [24, 25].

4. Materials and methods, lines 115-116. It is no clear how many spots have been used per strains and how many reads have been performed on each spot. 

  • Thank you for your comment. We have added the following information.

For each sample, at least eight mass spectra were collected through at least three repeat measurements; each measurement was consisted of 5 spots with at least 4 shots each.

5. In section 2.4: Have the authors used FlexAnalysis program to smoothing and baseline substraction of the spectra, or other analyses? If so, it should be mentioned. 

  • Yes, we did smoothing and baseline correction. We have added the description about that as follows in section 2.4.

Mass spectra were baseline corrected and processed smoothing with Biotyper v3.0 software (Bruker Daltonics).

6. Schizophyllum commune is already in the commercial database of Bruker. Have you evaluated the possibility of identify it with the manufacturer's database?

  • Thank you for your question. We compared our acquired S. commune's spectra with the information on S. commune in the commercial database in Biotyper 3.0, and the scores ranged from 1.0 to 1.7; the reason for that would be due to variation within the species or culture conditions or other factors. We added the description as follows in section 4.

Mass spectra of four strains of S. commune acquired in the present study were compared with the information on S. commune in the commercial database in Biotyper 3.0, and the scores ranged from 1.0 to 1.7; the reason for that would be due to variation within the species or to culture conditions or other factors.

7. The authors have evaluated the scores between raw spectra and MSP in the database for BjerkanderaPleurotusand Trametes. Why you have not included the other genera (FomitopsisSerpulaSchizophyllum)? It should be confirmed that they are also properly identified and no misidentifications at genus level are obtained. 

  • Thank you for your important suggestion. To enhance the database, intra- and interspecific variation should be examined in all fungal species. In the present study, mass spectra variation in three genera. We have added that as follows to Section 4.

To enhance the mass spectra database, intra- and interspecific variations should be examined in all fungal species. In the present study, mass spectra variation in three genera, Bjerkandera, Trametes, and Pleurotus, were investigated because each genus exhibited characteristic features of intra- and interspecific variation in the mass spectra: the genus Bjerkandera had two subgroups; in the genus Pleurotus, mass spectral analysis detected taxonomically ambiguous strains by DNA barcoding and suggested taxonomic position for the strains; and in the genus Trametes, mass spectra information could be clearly separated into species. The investigation suggested that mass spectra comparison is a good tool to identify and discriminate wood-rotting fungi.

Minor comments:

  1. Materials and methods: which was the approximate time that the cultures were incubated?
  • Thank you for your comment. The growth of basidiomycetes varies significantly in species and strains. Therefore, we used colonies that reached a stationary phase in this study. That has been described in 2.2, and also, examples of incubation periods have been given in 3.1

2. Figure 4, 5 and 6. In the legend, I recommend to change the score "≥1.7,<2.0" to "1.7-2.0" to be clearer. In addition, this box is lacking in the legend of Figure 5. 

  • Thank you for the comment. We corrected it.

3. Reference 33 have no title. Please, correct it. 

  • Thank you for the comment. We corrected it.

We would also like to thank reviewer 1  again for the opportunity to improve the manuscript with the reviewer's valuable comments and suggestions. We have made every effort to incorporate the reviewer's comments and hope that these revisions will make our submission acceptable.

Reviewer 2 Report

The authors have generated an in-house PMF database and differentiated between ten major species of wood-rotting fungi. Although this is not the first of this type of study, they have successfully differentiated between species and even between strains of the same species. Some minor queries and suggestions are requested-

11. How many times do each MSP were created?  

22. In figure-2, an outgroup can be added.

33. The authors are requested to clarify lines 180-181.

44. Have the authors compared the MSP with the existing database?

55. In some parts of the manuscript, the scientific names are not italicized. The authors are requested to edit the typos carefully. 

Author Response

Reviewer 2

We thank reviewer 2 for carefully reading our manuscript and providing numerous kind and helpful comments. We have revised the manuscript according to the reviewers' comments.

1. How many times do each MSP were created?  

  • Thank you for your suggestion. We have added the following information in section 2.3.

For each sample, at least eight mass spectra were collected through at least three repeat measurements; each measurement was consisted of 5 spots with at least 4 shots each.

2. In figure-2, an outgroup can be added.

  • Thank you for your comment. We corrected Figure 2.

3. The authors are requested to clarify lines 180-181.

  • Thank you for pointing out this v significant point. Thank you for pointing out this significant point. We have added a description of the examination of relationships within the genus Pleurotus (Fig. 5) in section 3.3.

Furthermore, examination within the genus Pleurotus based on mass spectral information (Figure 5) indicated that NBRC 8330 was closely related to P. pulmonarius NBRC 31345. Similarly, NBRC 33211 was shown to be closely related to P. eryngii. It has been suggested that these strains may be in a phylogenetically intermediate position of these species, respectively, and the analysis using mass spectra could represent that.

4. Have the authors compared the MSP with the existing database?

  • Thank you for your question. We compared our acquired S. commune's spectra with the information on S. commune in the commercial database in Biotyper 3.0, and the scores ranged from 1.0 to 1.7; the reason for that would be due to variation within the species or culture conditions or other factors. We added the description as follows in section 4.

Mass spectra of four strains of S. commune acquired in the present study were compared with the information on S. commune in the commercial database in Biotyper 3.0, and the scores ranged from 1.0 to 1.7; the reason for that would be due to variation within the species or to culture conditions or other factors.

5. In some parts of the manuscript, the scientific names are not italicized. The authors are requested to edit the typos carefully. 

  • Thank you for your comment. We corrected them.

We would also like to thank reviewer 2 again for the opportunity to improve the manuscript with the reviewer's valuable comments and suggestions. We have made every effort to incorporate the reviewer's comments and hope that these revisions will make our submission acceptable.

This manuscript is a resubmission of an earlier submission. The following is a list of the peer review reports and author responses from that submission.